Does silvoagropecuary landscape fragmentation affect the genetic diversity of the sigmodontine rodent Oligoryzomys longicaudatus?

Lazo-Cancino Daniela 1
Musleh Selim S. 2
Hernandez Cristian E. 1
Palma Eduardo 3
Rodriguez-Serrano Enrique enrodriguez@udec.cl 1
1 Departamento de Zoologia, Universidad de Concepción , Concepción , Biobío , Chile
2 Departamento de Oceanografía, Universidad de Concepción , Concepción , Biobío , Chile
3 Departamento de Ecología, Pontificia Universidad Católica de Chile , Santiago , Chile
Gandini Patricia
Electronic publication date: 2017 Sep 29
Publication date: 2017
Volume: 5
Electronic Location ID: e3842
Received 2017 Mar 29; Accepted 2017 Sep 1
Copyright: ©2017 Lazo-Cancino et al.
Copyright year: 2017
Copyright holder: Lazo-Cancino et al.
License: This is an open access article distributed under the terms of the Creative Commons Attribution License, which permits unrestricted use, distribution, reproduction and adaptation in any medium and for any purpose provided that it is properly attributed. For attribution, the original author(s), title, publication source (PeerJ) and either DOI or URL of the article must be cited.
License URL: https://creativecommons.org/licenses/by/4.0/

Keywords: Conservation genetics, Sigmodontinae, Genetic diversity, Genetic structure, Least cost path, Surrounding matrix

Funding: National Institute of Health ICIDR 1 U19 AI45452-01 DIUC 213.113.85-1AP FONDECYT 1130467–1170761 1140692–1170815 1170486 CONICYT Magister National scholarship 22150880 This work was supported by the following projects: “Hantavirus Ecology and Disease in Chile” grant National Institute of Health—ICIDR 1 U19 AI45452-01 (Eduardo Palma); DIUC grant 213.113.85-1AP (Enrique Rodríguez-Serrano); FONDECYT grants 1130467–1170761 (Eduardo Palma); 1140692–1170815 (Cristian E. Hernandez) and 1170486 (Enrique Rodríguez-Serrano), and by CONICYT Magister National scholarship, year 2015 - 22150880. The funders had no role in study design, data collection and analysis, decision to publish, or preparation of the manuscript.

==============================
Background

Fragmentation of native forests is a highly visible result of human land-use throughout the world. In this study, we evaluated the effects of landscape fragmentation and matrix features on the genetic diversity and structure of Oligoryzomys longicaudatus, the natural reservoir of Hantavirus in southern South America. We focused our work in the Valdivian Rainforest where human activities have produced strong change of natural habitats, with an important number of human cases of Hantavirus.

Methods

We sampled specimens of O. longicaudatus from five native forest patches surrounded by silvoagropecuary matrix from Panguipulli, Los Rios Region, Chile. Using the hypervariable domain I (mtDNA), we characterized the genetic diversity and evaluated the effect of fragmentation and landscape matrix on the genetic structure of O. longicaudatus. For the latter, we used three approaches: (i) Isolation by Distance (IBD) as null model, (ii) Least-cost Path (LCP) where genetic distances between patch pairs increase with cost-weighted distances, and (iii) Isolation by Resistance (IBR) where the resistance distance is the average number of steps that is needed to commute between the patches during a random walk.

Results

We found low values of nucleotide diversity (π) for the five patches surveyed, ranging from 0.012 to 0.015, revealing that the 73 sampled specimens of this study belong to two populations but with low values of genetic distance (γST) ranging from 0.022 to 0.099. Likewise, we found that there are no significant associations between genetic distance and geographic distance for IBD and IBR. However, we found for the LCP approach, a significant positive relationship (r = 0.737, p = 0.05), with shortest least-cost paths traced through native forest and arborescent shrublands.

Discussion

In this work we found that, at this reduced geographical scale, Oligoryzomys longicaudatus shows genetic signs of fragmentation. In addition, we found that connectivity between full growth native forest remnants is mediated by the presence of dense shrublands and native forest corridors. In this sense, our results are important because they show how native forest patches and associated routes act as source of vector species in silvoagropecuary landscape, increasing the infection risk on human population. This study is the first approach to understand the epidemiological spatial context of silvoagropecuary risk of Hantavirus emergence. Further studies are needed to elucidate the effects of landscape fragmentation in order to generate new predictive models based on vector intrinsic attributes and landscape features.

Introduction

Habitat fragmentation is widely recognized as a major threat to global biodiversity (Brooks et al., 2002; Secretariat of the Convention on Biological Diversity, 2005). In this process, a large wild habitat changes into a number of small isolated patches as consequence of human activities (Wilcove, McLellan & Dobson, 1986; Fahrig, 1997; Fahrig, 2003). Those changes imply gradual or accelerated reduction of original habitat’s area (Young, Boyle & Brown, 1996; Lande, 1998; Fahrig, 2003). The main intraspecific consequences of habitat fragmentation are discontinuities on the distribution of resources and species’ optimal environmental conditions, leading to a decrease in connectivity among fragmented populations (Vos & Stumpel, 1995; Lees & Peres, 2008). Thus, fragmentation isolates population, reduces gene flow, the genetic diversity and the effective population size, favoring genetic processes such as drift and inbreeding (Allendorf & Luikart, 2007; Johansson, Primmer & Merilä, 2007; Taylor et al., 2011). In addition, habitat fragmentation can affect the fitness reducing adaptive responses to local selective environments and may cause local extinction (Bolger et al., 1997; Frankham, 2005; Johansson, Primmer & Merilä, 2007; Willi et al., 2007; Bijlsma & Loeschcke, 2012).

Recently, it has been proposed that the structure of the landscape matrix is the main modulator of the consequences of fragmentation on biodiversity (Gascon et al., 1999; Debinski, 2006). Accordingly, if the surrounding matrix has structural similarity with the original habitat remnants, the inter-patch migration is granted avoiding important reduction of patch species richness (Gascon et al., 1999; Ricketts, 2001; Prugh et al., 2008; Franklin & Lindenmayer, 2009; Driscoll et al., 2013). Although this proposal has strong support on different fragmented systems, the genetic consequence of the fragmentation in vertebrates has a strong bias to Tropical Forest species (Radespiel & Bruford, 2014). Further, the intraspecific consequence of the matrix permeability has been studied mainly on European vertebrate or insect species (e.g., McRae, 2006; Arens et al., 2007; Emaresi et al., 2011; Van Strien, Keller & Holderegger, 2012).

The patch’s connectivity is often measured through fixation indexes and isolation by distance (Manel et al., 2003). However, Arens et al. (2007) use an explicit calculation of matrix permeability variables for the study of connectivity among Moor frog patches from Netherlands, highlighting the importance of incorporating the landscape complexity on the evaluation of genetic connectivity. This implies that a simple isolation by distance model is insufficient to explain the genetic diversity in a system of patches surrounded by a matrix of several land-uses. Thus, numerous approaches have been developed in recent years, accounting for the matrix complexity with different theoretical foundations. According to Van Strien, Keller & Holderegger (2012), these approaches can be categorized into two groups: those using transects and those using matrix features to establish landscape cost/resistance surfaces. For the latter, there are two popular approaches, least-cost distance model and the circuit theory model (e.g., Walker & Craighead, 1997; Adriaensen et al., 2003; McRae, 2006). The least-cost distance models minimize the travel distance among habitat patches and the cost traversed, offering the shortest cumulative cost-weighted distance (optimal route) between an origin patch to a destination patch. On the other hand, McRae (2006) proposed a model based on circuit theory that “predicts a positive relationship between genetic differentiation and the resistance distance, a distance metric that exploits precise relationships between random walk times and effective resistances in electronic networks”. Thus, these new approaches allow to understand the effecst on the genetic diversity of the patch-matrix dynamics.

During the last 30 years, the Mediterranean and the Temperate Chilean landscapes have been strongly modified by silvoagropecuary activities (agricultural, lumbering and industrial forestry activities), where remnants of native forests are restricted to zones with difficult access (Bustamante & Grez, 1995; Aguayo et al., 2009). Then, these patches of natural habitats constitute a highly fragmented environment, where patches are surrounded by different productive crops, exotic forestry species (mainly Eucalyptus globulus and Pinus radiata), and secondary regrowth native forest.

A frequent species in the Mediterranean and Temperate Forests of Chile is the long-tailed pygmy rice rat (“Colilargo”), Oligoryzomys longicaudatus (Bennett, 1832). This sigmodontine rodent has a broad distribution in Chile and Argentina. In Chile it occurs from 27°S to 54°S (Belmar-Lucero et al., 2009), whereas in Argentina ranges from 36°S to 51°S on the eastern slope of the Andes mountains (Carbajo & Pardiñas, 2007). Oligoryzomys longicaudatus shows a high vagility and a large home range 320–4,800 m2, with seasonal fluctuations (Murúa, González & Meserve, 1986). The species is mainly granivorous, inhabiting microhabitats with dense foliage, which could be related to its saltorial mode of locomotion and as a mechanism to avoid predators (Murúa, González & Jofre, 1980; Murúa & González, 1982). Molecular studies based on cytochrome b mitochondrial DNA (mtDNA) sequences have shown a marked genetic homogeneity along the species geographic distribution (Palma et al., 2005). However, studies based on hypervariable domain I (HVI) of the mtDNA recovered a geographical structure of populations in agreement with the ecoregions and the three recognized subspecies along the species range (Palma et al., 2012a), and a temporal genetic variability at local scale (Boric-Bargetto et al., 2012). In addition, O. longicaudatus has been the focus of numerous epidemiological studies, given that the species is the major reservoir of the Andes strain of Hantavirus that causes a cardiopulmonary syndrome to human populations with a mortality rate of about 35% (Toro et al., 1998; Martinez-Valdebenito et al., 2014). Thus, to evaluate the effects of changing landscapes on the migration and connectivity of O. longicaudatus populations, and the potential effect on infection rates, especially on peri-urban areas, constitute highly relevant issues to the ecology, the genetics and the epidemiology of this species (Torres-Pérez et al., 2004). Therefore, in this study we evaluated the effects of landscape fragmentation and matrix structure on the genetic diversity and genetic structure of O. longicaudatus. We focused our work on a portion of the southern Temperate Forest (the Valdivian Rain Forest) where human activities have produced a strong impact on natural habitats, and where an important number of human cases of Hantavirus have been reported in Chile (http://epi.minsal.cl/hantavirus-materiales-relacionados/).

Materials & Methods

Study site and specimens analyzed

The Valdivian Temperate Forest—southern Chile and nearby Argentina—is one of the 25 biodiversity hotspots of the world threatened by anthropogenic activities (Olson, 1998; Myers et al., 2000). This biome/ecoregion has been considered a biogeographic island that harbors a quite diverse assemblage of mammals where native species are mostly restricted to national parks and rural areas with fragmented landscapes (Echeverría et al., 2007). For this study, we took samples of O. longicaudatus from five different patches of dense native, full growth, temperate forests in a total area of 3.5 km2. The specimens were trapped with standard Sherman traps (8 × 9 × 23 cm; H. B. Sherman Traps, Inc., Tallahassee, FL, USA). The field trapping procedure was conducted through a regular grid sampling design, setting 450 traps night for three nights, and we used oat and vanilla as bait.

The study was conducted during the autumn and winter of 2007 in the locality of Curirruca, Panguipulli province, Los Rios region southern Chile (39°48′30″S, 73°14′30″W; Fig. 1). The captures were conducted under the Chilean Government authorization: Resolución Exenta No 7325 (December 30, 2005; from Servicio Agrícola y Ganadero, Ministerio de Agricultura, Gobierno de Chile). We selected this temporal window because it did not match with the reproductive period of the species and the mobility among patches should be reduced (Murúa, González & Meserve, 1986). The patches sampled were surrounded by a matrix—the rest of landscape after exclusion of habitat patches—characterized by recent adult and/or harvested plantations, grasslands and shrublands areas, agricultural fields, and/or adjacent to forest roads (Table S1). The appropriate landscape soil uses for the sampling year were obtained from the Chilean Government National Environmental Information System—SINIA (http://ide.mma.gob.cl/).

Figure 1 GIS representation and satellite image of the study site.

(A) Land uses, roads, and native forest patches (fragments “FRX”) where individuals of Oligoryzomys longicaudatus were sampled. (B) Google Earth image of the study site depicting the fragments and the high heterogeneity of the land uses. Map Data: Google Earth, DigitalGlobe.

The present study was conducted using blood samples and liver tissue from 73 specimens of O. longicaudatus, collected in this area (Table S2). All specimens were handled following the standard bioethical and biosafety protocols proposed by the American Society of Mammalogists (ASM; Sikes & Gannon, 2011), and the Center for Diseases Control and Prevention (CDC; Mills et al., 1995), respectively.

Laboratory methods

DNA was extracted using the Wizard® Genomic DNA Purification Kit (PROMEGA, Madison, WI, USA). Through the polymerase chain reaction (PCR) we amplified ∼1100 bp from mtDNA from which we used 527 bp corresponding to the hypervariable subunit I (HVI) and part of the conservative domain of the Control Region. The mammalian mtDNA hypervariable regions are included within the extended terminal associated sequences (ETAS) and conserved sequence block (CSB; Vigilant et al., 1991; Sbisà et al., 1997; Pesole et al., 1999). The evolution of D-loop region in mammals is characterized by a strong rate heterogeneity among sites, tandem repeated elements and high frequency of insertion/deletion events (Saccone, Pesole & Sbisà, 1991; Wakeley, 1993; Sbisà et al., 1997; Pesole et al., 1999). The ETAS and CSB domains evolve fast enough to be used for population genetics studies (Wakeley, 1993; Sbisà et al., 1997; Pesole et al., 1999; Rosel et al., 1999; Kerth, Mayer & König, 2000; Matson et al., 2000).

We used primers DLO-L (5′ CGG AGG CCA ACC AGT AGA 3′) and DLO-H (5′ TAA GGC CAG GAC CAA ACC 3′; Belmar-Lucero et al., 2009; Palma et al., 2012a) according to the following thermal profile: an initial denaturation of 5 min at 94 °C, followed by 25 or 30 cycles of denaturation for 30 s at 94 °C, annealing for 30 s at 57 °C and extension for 1 min 30 s at 72 °C, and a final extension of 5 min at 72 °C. The PCR products were sent to Macrogen (http://dna.macrogen.com/) for purification and sequencing (Applied Biosystems 3730XL sequencer; Applied Biosystems, Foster City, CA, USA). Sequences were edited with BioEdit v 7.2.5 (Hall, 1999) and aligned using ClustalW (Thompson, Higgins & Gibson, 1994).

Data analyses

Genetic diversity

To describe the genetic diversity in all the patches studied we used the DnaSP Software v 5.10.01 (Librado & Rozas, 2009) to estimate the number of haplotypes (Nh), segregating sites (S), the haplotype diversity (Hd) and the nucleotide diversity (π). The same software was used to calculate the GammaST (γST; Nei, 1982) a statistical index of genetic differentiation that represents an unbiased estimate of the population subdivision fixation index (FST) and its use is more appropriate for haplotype data. Statistical significance of genetic differentiation was tested using Hudson’s nearest neighbor statistics (Snn) with 1,000 permutations in DnaSP. Snn statistics indicates the frequency with which nearest neighbor sequences are found in the same group (Hudson, 2000).

Fragmentation effect

To evaluate the fragmentation effects on the genetic structure of O. longicaudatus we estimated the number of panmictic units in the landscape surveyed using the package GENELAND v 3.2.2 (Guillot, Mortier & Estoup, 2005) in the R software (R Core Team, 2016). For this, we follow the proposal of Guillot et al. (2012) codding the data in such a way that the various haplotypes of mtDNA are recoded as alleles of a single locus. This package implements a statistical model with Bayesian inference and uses geo-referenced data of the sequenced individuals, inferring and locating genetic discontinuities between populations. The number of clusters was determined by running MCMC (Markov chain Monte Carlo) iterations to estimate K (i.e., the most probable number of populations). The analysis was performed in both non-correlated and correlated model, allowing values of K to vary from 1 to 5, running MCMC with 10,000,000 iterations sampling each 1,000. To choose the best model that fits the data, a log10 Bayes Factor (BF) with 1,000 bootstrap replicates was performed in Tracer v 1.5 (http://tree.bio.ed.ac.uk/software/tracer/). The model that better fitted our data was the correlated allele frequency model, which assumes that rare alleles in a certain populations are also rare in other populations (Guillot, Mortier & Estoup, 2005).

Landscape matrix effect

For the identification of landscape matrix effects on the genetic structure of O. longicaudatus we followed three approaches. First, we performed isolation by distance test (IBD) as a null model, because this model contains no information about landscape features, where dispersal occurs in homogenous geographic spaces (Nowakowski et al., 2015). IBD was performed using vegan package 2.4-1 package (Oksanen et al., 2016) implemented in R (R Core Team, 2016). IBD was tested using a mantel test with 119 permutations between a matrix of genetic distances (γST) and a matrix of geographic distances between the five patches. Second, we used Least-Cost Path (LCP) analysis. In LCP, genetic distances between patch pairs increase with cost-weighted distances, taking into account the friction effects of the landscape on the individual movement process (Adriaensen et al., 2003; Epps et al., 2007). Third, we used isolation by resistance (IBR) analysis, where dispersal occurs in heterogeneous landscapes and the resistance distance is the average number of steps that is needed to commute between the patches during a random walk that is calculated using the circuit theory (McRae, 2006).

To estimate the distances under LCP and IBR models we used the package gdistance (Van Etten, 2017). For this, we first fed the gdistance package with a raster file containing the landscape features of the study area classified in five classes: native forest (all age classes), grassland and shrublands (with/out arborescent elements), farm (agricultural use), plantation forestry (monoculture of exotic species, eg. Pinus radiata) and mix forest (zone with both native and introduced trees). The raster resolution was 0.008 × 0.0006 pixels, but for technical feasibility we created a raster layer with larger cells (0.048 × 0.0034 pixels of resolution) using raster package 2.5-8 (Hijmans, 2015; Hijmans, 2016) in R (R Core Team, 2016). Second, we used transition function of gdistance to create a transitions matrix which represents the transition from one cell to another on a grid where each cell is connected to its 8 neightbours. In short, this function calculates the conductance values from the values of each pair of cells to be connected (Van Etten, 2017). However, there are two distance distortions that need to be corrected; diagonal neighbors are more remote from each other than orthogonal neighbours, and on a longitude-latitude grids, West–East connections are longer at the equator and shorter towards the poles. To solve these distortions, we used the geoCorrection function. For the transition matrix used in LCP, this function divides each value from the matrix by distance between cell centers (Van Etten, 2017). On the other hand, for the transition matrix used in IBR, the function weights the probability of reaching an adjacent cell in a random walk by making it proportional to the surface covered by the cell, multiplying the North–South transition values with the cosine of the average latitude of the two cell centers that are being connected (Van Etten, 2017). To calculate least-cost distances between patches, we used costDistance function that computes the cost units as the reciprocal of the values in the transition matrix using the Djkstra’s algorithm (Dijkstra, 1959). To calculate the resistance distances between patches, we used commuteDistance function, this function uses the algorithm implemented by Fouss et al. (2007) to calculate the expected random walk commute time between patches, resulting in the average number of steps needed to commute between the locations (Van Etten, 2017).

To perform the correlation between genetic distances (γST) with both, LCP and IBR distances, we used vegan package 2.4-1 package (Oksanen et al., 2016) in R (R Core Team, 2016) performing a mantel test with 119 permutations. This number of permutations is the maximum number allowed to avoid duplication due the size of matrix distance (i.e., 5 × 5). Finally, we used gdistance 1.1-9 package (Van Etten, 2015) to trace the quickest path among pairs of patches for LCP model applying shortestPath function which calculates the shortest path from one patch to another. This allowed us to know if there were specific types of soil use (the surrounding matrix) that the “Colilargo” uses as a corridor.

Results

Genetic diversity

We found a total of 43 segregating sites and 36 haplotypes out from 73 sequences (Table 1). All patches showed a high haplotype diversity (Hd), although FR5 showed the lowest value (0.867) whereas FR2 exhibited the highest values (0.987) (Table 1). Regarding nucleotide diversity (π), it was low, ranging from 0.012 (FR4) to 0.015 (FR1) (Table 1).

Table 1 Descriptive statistics of the genetic variation of O. longicaudatus sequences for each sampled patch.

Patches	N	S	Nh	Hd ± SD	π± SD	
FR1	12	22	9	0.939 ± 0.058	0.015 ± 0.002	
FR2	18	33	16	0.987 ± 0.023	0.014 ± 0.001	
FR3	9	20	8	0.972 ± 0.064	0.013 ± 0.002	
FR4	24	28	13	0.880 ± 0.056	0.012 ± 0.002	
FR5	10	21	7	0.867 ± 0.107	0.013 ± 0.002	
Total	73	43	36	0.954 ± 0.012	0.014 ± 0.001	
Notes.

N Number of individuals

S segregating sites

Nh haplotype number

Hd haplotype diversity

π nucleotide diversity

SD standard deviation

Genetic structure

Results of GENELAND v 3.2.2 (Fig. 2) for population genetic structure inferred that the most probable number of clusters of individuals was two (K = 2). The first cluster grouped sequences from FR3 and FR4 (Fig. 2A). The second cluster joined the patches FR1, FR2 and FR5 (Fig. 2B). We found high posterior probabilities (0.9) for cluster assignation (Fig. 2).

Figure 2 Spatial population structure.

GENELAND analyses with posterior probability isoclines denoting the extent of genetic landscapes. Black dots represent patches analyzed. White indicates regions with the greatest posterior probability of inclusion, whereas diminishing probabilities of inclusion are proportional to the degree of coloring. (A) Map of posterior probability to belong to cluster 1; (B) map of posterior probability to belong to cluster 2.

Pairwise γST values (Table 2) revealed the occurrence of genetic differentiation recorded for FR3 and FR5 (γST = 0.099, p = 0.046), and between FR4 and FR5 (γST = 0.085, p = 0.044). The remaining patches showed γST values ranging from 0.022 to 0.096 (not significantly different from 0).

Landscape matrix effects

We found that O. longicaudatus did not exhibit significant isolation-by-distance, γST was not correlated with geographic distance (r = 0.694, p = 0.075, Fig. 3A). However, we found for the least cost path approach a significant positive relationship (r = 0.737, p = 0.05, Fig. 3B). But, the long-tailed pygmy rice rat did not exhibit a significant isolation by resistance relationship for genetic differentiation (r = 0.740, p = 0.058, Fig. 3C). The shortest paths traced for LCP shows that individuals of O. longicaudatus moved with preference through young and all growth native forest, and grassland with arborescent shrublands to connect patch pairs (Fig. 4).

Table 2 Pairwise genetic distances between patches (γst) (below diagonal) and SNN significance P-values (above diagonal).

Bold numbers represent significant values.

	FR1	FR2	FR3	FR4	FR5	
FR1	–	0.835	0.735	0.588	0.31	
FR2	0.022	–	0.586	0.351	0.379	
FR3	0.091	0.049	–	0.346	0.046	
FR4	0.096	0.062	0.030	–	0.044	
FR5	0.038	0.027	0.099	0.085	–	

Figure 3 Effect of fragmentation and landscape matrix on the genetic structure of O. longicaudatus.

Graphics of Pearson correlation coefficient (r). (A) Isolation by Distance (IBD), (B) The Least-cost Path (LCP), and (C) Isolation by Resistance (IBR). r value corresponds to Pearson correlation coefficient and p values correspond to significance (p ≤ 0.05).

Figure 4 Results of Least Cost Paths analyses.

Satellite image of the study site depicting the fragments surveyed and the least cost paths inferred for Oligoryzomys longicaudatus in the highly fragmented Valdivian Rainforest of southern Chile. Google Earth image © 2017 DigitalGlobe.

Discussion

In this work, we found that, at this reduced geographical scale, Oligoryzomys longicaudatus shows genetic signs of fragmentation. Also, we found that genetic distance between patches showed best fitting to a LCP model. In addition, we found that connectivity between full growth native forest remnants is mediated by the presence of dense shrublands and native forest corridors. The latter can be composed by different age and health status native formations (Fig. 1B; Fig. 4). These results complement previous efforts to understand the association between landscape attributes and the genetic diversity of this species. Previous studies have been focused on regional and local scales proposing relationships between haplogroups and ecogeographic regions, as well as latitudinal genetic structure in local context (Belmar-Lucero et al., 2009; Palma et al., 2012a; Ortiz et al., 2017). Specifically, Ortiz et al. (2017), studying a fragmented landscape in the Argentinian Patagonia, found that landscape features such as lakes, rivers, roads and urban settlements constrain the movement of O. longicaudatus, acting as barriers reducing gene flow. Our results support the latter findings since, even at this small scale, showing abrupt changes in land use being the species strongly affected by the fragmentation of the primary habitat.

The long-tailed pygmy rice rat has a marked foraging behavior characterized by the search of seeds, a highly localized and temporally variable resource (Murúa, González & Meserve, 1986). Previous studies on this species suggested high flexibility in habitat use, characterized by an opportunistic behavior and large home range (Murúa & González, 1986; Murúa, González & Meserve, 1986; Spotorno, Palma & Valladares, 2000). In addition, studies based in the Valdivian Rainforest (dense full growth forest at Villarrica National Park) strongly suggest that migration is the modulator of the diversity and temporal genetic structure of O. longicaudatus (Boric-Bargetto et al., 2012). Thus, the landscape matrix effects on the genetic diversity and genetic structure of this species would be largely buffered by its high vagility features particularly in the Valdivian Rainforest which is its primary habitat (Murúa, González & Meserve, 1986). However, we found that the LCP model is the best predictor of the genetic distance for the fragments surveyed. This implies that O. longicaudatus minimize the tradeoff between distance travelled and the costs traversed. Graphically, our results showed that the shortest paths among patch pairs are across native forest (the species’ primary habitat), representing the routes of maximum efficiency for landscape connectivity, while the rest of land uses would act, in some extent, as barriers.

Our results should be viewed with caution, in terms of some possible consequences derived from our interpretations. If the most efficient routes connecting patches are through native forest, this does not mean that individuals of O. longicaudatus occurs only in native forest. If dispersing individuals follow these optimal routes, they would increase the probability of survival reaching the optimal destination (another patch). This is because, dispersal through optimal routes and their associated habitats increases increases the likelihood of finding resources and evading predation (Walker & Craighead, 1997). In fact, Moreira-Arce et al. (2015) found that O. longicaudatus is the preferred prey of the culpeo fox (Lycalopex culpaeus) only in monocultures, increasing its predation risk in silvoagropecuary landscapes. However, dispersing individuals may not choose the optimal route, and travel through other types of soil use less effectively, because of connectivity among patches. For instance, due to its opportunistic behavior and according to previous ecological studies settled in coastal ranges of the Valdivian Rainforest, this species could be found on grassland—shrublands when the seed availability increases on this area but not on the native forest (Murúa & González, 1986; Murúa, González & Meserve, 1986). In tree monocultures, this rodent is less abundant, but not absent, than in native forest and exhibits an omnivorous diet, where mainly consumes seeds (e.g., Pinus radiata seeds) and fruits, and arthropods and mushrooms in less amount (Muñoz-Pedreros, Murúa & González, 1990; Moreira-Arce et al., 2015). Another important caveat of this work is the molecular marker used to infer the genetic structure and diversity of the Colilargo. We used a very variable fragment from the mtDNA, then our results reflect the matrilineal genetic diversity and least cost paths. The major consequence of this choice, is that just a quarter of the total effective population size is used in this study, so the results could underestimate the genetic diversity of each patch, but not the routes, since the latter are estimated from the properties of the landscape (Van Etten, 2017). Finally, given that our sampling period was during the autumn and winter, our results may reflect connectivity aspects of Colilargo modulated by the features of those seasons. Interestingly, a previous ecological work on O. longicaudatus, settled in Temperate Forest, shows that a very marked seasonal reproductive period from October to May, which overlaps with the recruitment period from March to April, is followed by population peaks during autumn-winter (Murúa, González & Meserve, 1986). Therefore, our sampling was carried out post-recruitment, so the results of genetic diversity and structuring are relevant to estimate connectivity in this fragmented system since these patterns are the result of migration processes between patches that have already occurred.

Oligoryzomys longicaudatus is recognized as the major reservoir of the Andes strain of Hantavirus (ANDV) in Southern South America (Medina et al., 2009). This virus causes the Hantavirus Cardiopulmonary Syndrome (HCPS) disease (Martinez-Valdebenito et al., 2014). In Valdivia, Chile, Mansilla (2006) found that >50% of HCPS cases were associated to silvoagropecuary landscapes (Holz & Palma, 2012). In addition, the long - tailed pygmy rice rat is one of the most common species in the rodent assemblages, where it could potentially infect other wild rodent species with the ANDV (horizontal transmission to coexisting species; Palma et al., 2012b), thus increasing the risk to humans (Polop et al., 2010; Andreo et al., 2012; Barrera & Murúa, 2016). In this sense, our results are important because they show how native forest patches and associated routes act as source of vector species in silvoagropecuary landscape, highly associated to human activities increasing the infection risk on human population. Further studies are required to elucidate the effects of landscape fragmentation at large scales (i.e., ecogeographic), in order to gain a deeper understanding of the underlying causes of HCPS infection risk in the Valdivian Forest. Finally, a general pattern of the consequences of Temperate Forest fragmentation should be based on an important number of species, and future efforts should point out to other endemic species of this ancient landscape of South America.

Supplemental Information

Table S1 Criteria for delimitation of patch areas

Criteria for delimitation of patch areas. Criteria were designed visualizing the patches in Google Earth Pro v 7.1 (http://www.google.com/earth/) and based on Soil Use Cover data of the Chilean National Environmental Information System (http://ide.mma.gob.cl/).

Click here for additional data file.

Table S2 Data for each individual used in this work

Sampling site, patch name, coordinates of specimens, GenBank accession number and voucher number of each specimen used in this study.

Click here for additional data file.

Supplemental Information 1 Raster File for landscape features

Raster file with landscape soil uses for 2007 at the study site with 0.048 × 0.0034 pixels of resolution.

Click here for additional data file.

Supplemental Information 2 Gamma_st distances_

Genetic distances between patches.

Click here for additional data file.

Supplemental Information 3 Georreferences for each patch

Georreferences (decimal system) for each patch surveyed.

Click here for additional data file.

Supplemental Information 4 Code to perform the analyzes used in this study

Annotated R code. This should be used in combination with the raw data delivered in supplementary materials: “comb_999.grd.zip”, “gammast.txt”, and “lonlat.txt”.

Click here for additional data file.

We thank the Laboratory work of RA Cancino and the Acuigen Team (R Galleguillos, CB Canales-Aguirre, and S Ferrada). In addition, we are deeply grateful to the Chilean Hanta Field Team in Panguipulli (S Belmar, G Carrasco, V Castro, V Escobar, P Gutiérrez, and R Thompson). Finally, we acknowledge all comments and suggestions made by the Academic Editor and three anonymous reviewers.

Additional Information and Declarations

Competing Interests

Author Contributions

Animal Ethics

DNA Deposition

Data Availability

The authors declare there are no competing interests.

Daniela Lazo-Cancino conceived and designed the experiments, performed the experiments, analyzed the data, contributed reagents/materials/analysis tools, wrote the paper, prepared figures and/or tables, reviewed drafts of the paper.

Selim S. Musleh performed the experiments, analyzed the data, contributed reagents/materials/analysis tools, wrote the paper, prepared figures and/or tables, reviewed drafts of the paper.

Cristian E. Hernandez analyzed the data, contributed reagents/materials/analysis tools, wrote the paper, reviewed drafts of the paper.

Eduardo Palma performed the experiments, analyzed the data, contributed reagents/materials/analysis tools, wrote the paper, reviewed drafts of the paper.

Enrique Rodriguez-Serrano conceived and designed the experiments, analyzed the data, contributed reagents/materials/analysis tools, wrote the paper, reviewed drafts of the paper.

The following information was supplied relating to ethical approvals (i.e., approving body and any reference numbers):

The capture was done under authorization from the Servicio Agrícola y Ganadero, Ministerio de Agricultura, Gobierno de Chile.

The following information was supplied regarding the deposition of DNA sequences:

The Hipervariable Domain I, of the Control Region (mtDNA), sequences used here are available via GenBank accession numbers KY211763 to KY211766, KY211783 to KY211818 and KY211820 to KY211862.

The following information was supplied regarding data availability:

The raw data has been supplied as Supplemental Files.

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
