# Peer review of "Does silvoagropecuary landscape fragmentation affect the genetic diversity of the sigmodontine rodent Oligoryzomys longicaudatus?"

_PeerJ, doi:10.7717/peerj.3842_

## Round 0.1 · original submission · Major Revisions

The subject of the article is interesting and could provide valuable information but needs some modifications in order to achieve validity in its results and in the interpretation of the data.

The results provided are not original because the use of shrubs and grassland by O. longicaudatus already been extensively described. Therefore, to evaluate the effects of landscape fragmentation on genetic diversity, and to investigate the most probable landscape characteristics of commuting “least-cost corridors” among patches give more interest for their publication. To integrate this information with the structure of the landscape is novel for both the species and the study area. I consider that from the suggestions provided and making a good integration of the results in the discussion the article can improve its quality in order to be considered for publication. Also, the fact that the studied species is a reservoir of the Hantavirus emphasizes the importance of the results for regional sanitary controls, in order to reduce the risk of humans to contract HPS.
Your paper has been evaluated by three peer reviewers, two of them specialist in genetics and ecology of your studies’ species. The three of them all are quite critical, express major concerns and all of them, touch not only shortcomings in the presentation of your work (starting with the need of a thorough language makeover) but they also provide some general comments on more severe issues. When revising your ms, I sincerely ask you to adequately address EACH of the referees' comments, as well as EACH of my general remarks on format issues, by either incorporating the suggestions in the revision, if possible, or providing brief but convincing rebuttals in case you do not agree with them.

Mayor concerns are: Rev 3, thought methodology was not properly chosen, but all of them agreed you need to clarify interpretation of the 2 clusters and to reformulate Figures 4A and 4 B. Although, methods have been described with sufficient information to be reproducible by other investigators, criteria used to characterize the landscape structured needs to be clarified. Reviewers also have some doubts about the interpretation of the data. There are more suggestions and concerns in Reviewers comments that surely will improve your Ms.

Reviewer 1 ·

Basic reporting

No particular comments.
There are minor grammatical problems throughout the text that will require minimal corrections. One issue is a need for additional punctuation (commas) to divide sentences.

Experimental design

The study focuses on the effect of landscape fragmentation on the genetic structure of Oligoryzomys longicaudatus that acts as a main reservoir of Hantavirus in South America. They realized their sampling at a small scale (3.5km2) in a region constituted of different landscapes. One important concern of this study is the type of molecular marker that authors used (mtDNA). Even if they argue that, this marker can be used for population genetics studies, at this geographical scale, I am not confident about the obtained results. This represent a problem since other confounding factors can also have a major impact (such as demography, differences between male and female). Microsatellites markers should have been used to answer to this question. The authors also chose to sample Oligoryzomys populations during one season (during the period of low mobility). It would have been interesting to compare their results with the period of high mobility to see how landscape influences the genetic structure of this species.

Validity of the findings

The results of landscape analysis could be given in more details (explaining clearly figure 4). As isolation by resistance analysis is not significant why comparing it to least-cost path results.
Authors should discuss about the factors that cannot be identified using mitochondrial markers as it is more adapted to large scale studies and long evolutionary process.

Additional comments

In their manuscript, the authors highlight the importance of landscape genetic studies of reservoir species in order to better understand the viral circulation and risk of emergence. The purpose of this study is really interesting and could correspond to the scope of the "PeerJ".
A more detailed justification for the results and addition of those obtained by Ortiz (2016) would improve the manuscript.
The authors could add reference: Ortiz N, Polop FJ, Andreo VC, Provensal MC, Polop JJ, Gardenal CN, Gonzalez-Ittig RE. 2016. Genetic population structure of the long-tailed pygmy rice rat (Rodentia, Cricetidae) at different geographic scales in the Argentinean Patagonia. Journal of Zoology.
In the abstract :
Line 47: replace “hanta” by “hantavirus” and “contagion” line 49 by “emergence”.

Reviewer 2 ·

Basic reporting

The objective of this paper is to evaluate the effects of landscape fragmentation and matrix features on the genetic diversity and structure of Oligoryzomys longicaudatus, a sigmodontine rodent species, natural reservoir of Hantavirus in southern South America. The writing is simple and easy to understand. The theoretical framework is properly formulated and framed on the centrality of the issue. The references are appropriate and updated. More than 45% of the references correspond to the last 10 years. The article is correctly structured in its organization. The tables are self-explanatory and provide appropriate information. The figures are pertinent but I suggest that the metric scale be incorporated in Figure 1. This will allow a better interpretation of the distance relationship between the sampling sites (patches) of O. longicaudatus. The results obtained in the study, although they contribute to describe the use of the environment by O. longicaudatus through the use of corridors that communicate the patches with the native forest, these have already been described in the bibliography. The most interesting contribution of the article is its methodology focused on the use of 3 distances (IBD, LCP and IBR) to describe the effect of the landscape matrix on the genetic structure of O. longicaudatus.

Experimental design

The experimental design is appropriate and the questions are addressed with original methodologies. The identification of the effect of the landscape matrix on the genetic structure of O. longicaudatus allows to evaluate the effect of landscape changes on the migration and connectivity of these populations of high epidemiological importance. Although the methodology in general is well explained I suggest that it is necessary to go into more detail in the description of the structure of the landscape. It is understood from the map legend that land use is classified into 6 different types of coverage but missing elements to interpret the layout of distances depending on each model. The methodology describes in detail the statistical package used and, from the technical point of view, the solution to certain difficulties in determining the distances. However, it is not clear how the environmental variables of each capture location are weighted in the sampling grid and how the data matrix was constructed to determine the different types of distances. Therefore, I suggest that the explanation be extended at this point in order to achieve a better interpretation of the results.

Validity of the findings

The results presented indicate that O. longicaudatus shows high haplotype diversity, but a general low nucleotide diversity, suggesting recent differentiation and potential changes in population size. Although these findings are robust, it is not clear the interpretation of the differentiation in the genetic structure of the 2 clusters as show the significant LCP analysis. It is not interpreted from Figure 4A that the shortest paths traced for LCP are through grassland /shrubland to connect patches of native forest. The same difficulty arises from FIG. 4B in which it is not interpreted that the shortest paths by means of the IBR are that individuals are moving through native forest only, but through longest path. I suggest improving Figure 4 so that the differences between the traces obtained by the 2 approaches (LCP and IBR) can be more clearly observed. I would also recommend that data of average distances (or other variables) be included to justify in each case the alternative use that the species makes of the landscape elements, either from the matrix or from the corridors.
On the other hand, the authors interpret that the use that would make the species of grassland/shrubland as corridors could be explained from their diet. However, the captures were made in winter and in this time of the year the availability of seeds is limited by the plant's own phenology.
The authors conclude that, taking into account the effects of the surrounding matrix, O. longicaudatus individuals would preferably use grassland / shrubland to connect native forest patches. Although the conclusion is well established and relates to the original question I find certain limitations since it is not clear the interpretation of the results obtained from the analysis of the distance of least cost, on which it is based.

Additional comments

The subject matter is of interest in the area of ecoepidemiology since it contributes to understand the spatial context of silvoagropecuary risk of Hantavirus contagion. Although it presents clarity in the objective of its investigation and good articulation between the conceptual framework and the development of its different sections, I consider that the results are partially supported by the data. I suggest that the article needs to specify, in more detail, the criteria used to characterize the landscape structure, to reformulate Figures 4 (A and B) in order to show differences in LCP and IBR tracings and to provide average distance data in order to interpret the differential use of the environment according to both analyzes. I believe that these modifications could contribute by giving greater reliability and validity of the results and to a greater support of the discussion.

Reviewer 3 ·

Basic reporting

The manuscript is well written, and structured, but, the English language should be improved.. Literature references are well selected. Enough background is given in relation to the problem being addressed. All appropriate raw data have been made available.

Experimental design

This paper reports a landscape genetic study of the sigmodontine Oligoryzomis longicaudatus (“Colilargo”) based on the analysis of 5 populations sampled in 5 patches located in a fragmented landscape with different types of soil use. The aim of this study is to evaluate the effects of landscape fragmentation on genetic diversity, and to investigate the most probable landscape characteristics of commuting “least-cost corridors” among patches The research question is well defined. The fact that the studied species is a reservoir of the Hantavirus emphasizes the importance of the results for regional sanitary controls, in order to reduce the risk of humans to contract HPS. However, data are neither robust enough nor statistically sound. The main conclusion drawn from the least-cost corridors used by this species is confusing because the results are not clearly presented. Besides, the statistical analysis should be improved.
Methods have been described with sufficient information to be reproducible by other investigators. However, the methodology used to describe geographic distribution patterns of genetic variation (i.e. GENELAND) was not appropriately chosen. Geneland is a statistical computer program for population genetics data analysis. Its main goal is to detect population structure in the form of systematic variation in allele frequency by departure from Hardy-Weinberg and linkage equilibrium. Geneland requires individual multilocus genetic data that are optionally geo-referenced. It implements several models that use both geographic and genetic information to estimate the number of populations in a dataset and delineate their spatial organization (Guillot et al., 2005). In the present study, the authors use a single molecular marker, instead of multilocus data, thus preventing the assessment of allelic frequencies, which are necessary to detect departures from HW equilibrium. Furthermore, if each nucleotide position were considered as a different locus, then all positions would be linked and not in linkage equilibrium. This analysis should not have been performed with only one sequence marker.

Validity of the findings

With respect to the validity of the conclusions I have some doubts about the interpretation of the data. They authors inferred correlations between geographic distances (calculated as IBD, LCP and IBR distances) with genetic distances through Mantel tests. They found that the only significant correlation was “genetic distance vs. least-coast distance”, and then, they calculated the shortest paths based on LCP and IBR. Based on these analyses, they concluded that the shortest path traced for LCP and IBR indicate that individuals of Oligorizomis are moving through “grassland/shrubland” and “native forests”, respectively, to connect patches of native forests. However, Figures 4 A and B do not show this difference. Instead, both figures show very similar results, with quicker paths occurring either across plantation forestries or native forests. This conclusion should be supported by providing better explanations and figures.

The authors apply the raster of land use only to show its distribution on a map with the least cost path superimposed. They do not use this variable to analyze the landscape resistance as a function of dispersal probabilities. Perhaps it would be appropriate to perform some kind of simulation analysis (i.e. CDPOP, Landguth and Cushman, 2010) that takes into account landscape heterogeneity, to see if the pattern of distribution of genetic variability obtained with empirical studies, is similar to that recovered from simulated data.

Additional comments

No comments.

---

## Round 0.2 · Minor Revisions

The paper improved considerably. Only some minor revisions have been suggested by one of the reviewers. Take special attention that the data has been taken in a given period of the year, so it is necessary to discuss your results within this context. Different results can be observed if you sample during other period of the season. I hope you can make this changes soon and resubmit the paper.

Reviewer 2 ·

Basic reporting

The manuscript improved significantly its quality from the changes made. The incorporation suggested in figure 1 of the metric scale as well as the palette of colors and the image in google earth undoubtedly facilitates the identification of the different environments and their interpretation.Likewise, the division into sections of data analysis orders its reading and facilitates its interpretation satisfactorily

Experimental design

The modifications made improve the interpretation of the results. The methodology was described in greater detail facilitating greater understanding and interpretation of the analyzes.

Validity of the findings

In general terms, the explanations included in this new version are consistent with the conclusions reached by the authors. The results have been discussed in greater depth and also, the included bibliography contributes satisfactorily to improve the quality of the manuscript. However, since the data were taken in a given period of the year it is necessary to be discussed within that context. Therefore it should be clarified in the discussion that the interpretation that is being made corresponds to the period of less mobility of O. lonficaudatus since in the reproductive season similar results could not necessarily be observed.

Additional comments

No comment.

Reviewer 3 ·

Basic reporting

This paper reports a landscape genetic study of the sigmodontine Oligoryzomis longicaudatus (“Colilargo”) based on the analysis of 5 populations sampled in 5 patches located in a fragmented landscape with different types of soil use. The aim of this study is to evaluate the effects of landscape fragmentation on genetic diversity, and to investigate the most probable landscape characteristics of commuting “least-cost corridors” among patches The research question is well defined.

Experimental design

Although a single marker was included in the analysis, it shows enough variability as to reveal patterns of distribution of variability in the landscape. Geneland is not the best approach for this analysis. Authors would have to justify their inclusion, describing that mitochondrial sequences were treated as a single loci, with many allelles (i.e. haplotypes).

Validity of the findings

The fact that the studied species is a reservoir of the Hantavirus emphasizes the importance of the results for regional sanitary controls, in order to reduce the risk of humans to contract HPS

Additional comments

All suggestions made by the reviewers were properly adressed. Therefore, I have no further comments.

---

## Round 0.3 · accepted · Accept

Authors have included all reviewers suggestions